# Runs of Homozygosity Preliminary Investigation in Pig Breeds

**DOI:** 10.3390/ani15070988

**Published:** 2025-03-29

**Authors:** Yuqiang Liu, Guangzhen Li, Wondossen Ayalew, Zhanming Zhong, Xiaohong Liu, Jiajie Sun, Jiaqi Li

**Affiliations:** 1State Key Laboratory of Swine and Poultry Breeding Industry, National Engineering Research Center for Breeding Swine Industry, Guangdong Provincial Key Lab of Agro-Animal Genomics and Molecular Breeding, College of Animal Science, South China Agricultural University, Guangzhou 510642, China; yuqiangliu123@126.com (Y.L.); guangzhenli6228@126.com (G.L.); wondessenayalew9@gmail.com (W.A.); zhongzhanming204@163.com (Z.Z.); jiajiesun@scau.edu.cn (J.S.); 2State Key Laboratory of Biocontrol, School of Life Sciences, Sun Yat-sen University, Guangzhou 510275, China; xhliu@163.com

**Keywords:** breeding strategies, crossbreeding effects, domestication, genetic diversity

## Abstract

This study investigated genetic diversity in domestic and wild pig populations across Asia and Europe by analyzing patterns of genomic regions with identical DNA segments inherited from both parents. These regions, often linked to inbreeding and selective breeding, were more frequent and longer in European pigs, reflecting intensive modern breeding practices. Asian pigs showed shorter segments, suggesting ancient inbreeding. Crossbreeding reduced these regions, improving genetic diversity in hybrid pigs. Key genomic areas under strong selection were associated with traits like disease resistance, growth, and reproduction. For example, a gene linked to growth rates in European pigs showed distinct genetic patterns compared to Asian pigs. These findings help farmers and breeders improve pig health and productivity by balancing genetic diversity with selective breeding, supporting sustainable livestock practices and food security. This work highlights the importance of genetic monitoring to maintain resilient pig populations while meeting global demands for animal protein.

## 1. Introduction

In diploid organisms, runs of homozygosity (ROH) serve as key markers of population genetics, shaped by natural selection, artificial selection, population structure, and inbreeding. Since Broman and Weber (1999) first described long homozygous segments in the human genome [1], ROH analyses have expanded to livestock species, including cattle [2,3], pigs [4,5], horses [6], sheep [7], and goats [8], offering insights into demographic history, genetic relationships, inbreeding levels, and selection effects.

Pigs have undergone distinct domestication histories in Asia and Europe, resulting in unique genetic architectures. Previous studies have shown that Asian domestic pigs predominantly harbor long ROH segments, reflecting recent bottlenecks. In contrast, European pigs exhibit more numerous but shorter ROH segments, indicative of intensive artificial selection [5]. This continental divergence provides a unique model system to investigate how human-mediated selection pressures differentially sculpt genome architecture during breed formation. In commercial breeds, ROH islands are enriched for genes linked to economically important traits, such as reproduction in Yorkshire pigs [9], immunity and growth in Duroc pigs [10], and fat deposition in Large White pigs [11]. Identification of candidate genes in commercial pig breeding populations will be applied to pig breeding to improve pig economics. Conversely, ROH in indigenous breeds are often associated with environmental adaptability, helping breeders develop more effective conservation programs and breeding materials for resilience [12,13,14].

Despite significant advances, inconsistencies in ROH detection methodologies complicate cross-study comparisons. Differences in ROH length thresholds, allowances for heterozygous genotypes, single-nucleotide polymorphism (SNP) density, and sequencing depth can affect results [15]. Commonly used detection tools, including Plink [16], Germline [17], and Beagle [18], employ varying algorithms, with Plink being the most widely applied.

In this study, we leveraged whole-genome resequencing data from 1203 pigs to systematically investigate the global distribution of ROH. Using standardized parameters in PLINK, we characterized differences in ROH between Asian and European pig populations and assessed the impact of hybridization. Our findings reveal fundamental differences in ROH patterns, highlight breed-specific selection pressures, and offer insights into genetic diversity and breeding strategies.

## 2. Materials and Methods

### 2.1. Population and Genomic Data

Whole-genome resequencing data were obtained from the pig genomics reference panel (PGRP, V1) [19]. Initially, we removed 45 samples from closely related Sus species in the PGRP panel. Subsequently, 374 samples with ambiguous ancestry were excluded, followed by the removal of 144 samples representing breeds with fewer than five individuals. Ultimately, 1203 individuals remained for analysis. SNP data were filtered using Plink (v1.90), removing variants with a minor allele frequency (MAF) < 0.01 or a missing genotype rate > 0.1 [11]. All autosomes were kept, and sex chromosomes were removed. After a serious filtration step, a total of 33,623,520 SNPs were retained to perform association analysis.

### 2.2. ROH Detection

ROH were identified using Plink (v1.9) with the following criteria: (i) sliding window of 50 SNPs; (ii) minimum overlap of 0.05 between consecutive windows; (iii) minimum of 100 consecutive SNPs per ROH; (iv) SNP density of at least one per 50 kb; (v) minimum ROH length of 500 kb, with a maximum gap of 100 kb between consecutive homozygous SNPs; and (vi) allowance for up to two missing genotypes and one heterozygous genotype per ROH segment.

ROH segments were categorized into three classes based on length: short (0.5–2 Mb), medium (2–5 Mb), and long (>5 Mb). For each population, we first calculated the total number and cumulative length of ROH within each category. Subsequently, these values were normalized by dividing the total counts and lengths by the number of individuals in the population, yielding population-level averages for both ROH counts and segment lengths per individual.

### 2.3. Genetic Differentiation and Selection Signatures

Pairwise F*_ST_* values between hybrid and parental populations were computed using the hierfstat package (v0.5-11) in R (V4.4.2). SNP filtering was conducted using Plink (v1.9) “--indep-pairwise 50 5 0.5”, retaining 7,897,020 SNPs. F*_ST_* values were calculated via hierfstat and visualized with ComplexHeatmap (v2.22.0).

To evaluate the differences in selection pressure between ROH islands and non-island regions, we defined ROH islands as genomic regions overlapping with the top 1% of high-frequency ROHs (threshold determined by ranking all ROHs based on their population carrier rates) and extracted SNPs within these regions. An equivalent number of SNPs were randomly sampled from non-ROH island regions as a control set to calculate selection signatures. We employed multiple complementary approaches to assess selection pressure: F*_ST_* (measuring population differentiation), Tajima’s D (detecting deviations from neutral polymorphism distributions), XP-EHH (comparing haplotype extension across populations), and π (nucleotide diversity). These methods capture selection signals through distinct perspectives: F*_ST_* identifies regions of divergent selection between populations, Tajima’s D distinguishes balancing selection (positive values) from directional selection (negative values), π quantifies diversity loss under recent selective sweeps, and XP-EHH detects population-specific positive selection. Their combined application enables cross-validation of strong selection signals in ROH islands while mitigating method-specific biases. Nucleotide diversity (π), F*_ST_* and Tajima’s D were calculated using VCFtools (v0.1.17) [20] with a 2-kb window. XP-EHH were calculated usingselscan (v2.0) [21] and F*_ST_* were implemented to evaluate the selection intensity of European and Asian populations. We defined ROH islands and annotated these regions using Ensembl (*Sscrofa11.1* v100). Genes located within ROH island regions were considered candidate genes. Gene Ontology (GO) and Kyoto Encyclopedia of Genes and Genomes (KEGG) enrichment analyses were conducted using clusterProfiler (v4.0) [22] to infer the biological functions of candidate genes.

## 3. Results

### 3.1. ROH Distribution Across Global Pig Populations

After stringent filtering, 1203 individuals across 49 breeds remained (Appendix A), with 3,485,897 high-quality SNPs retained for downstream analysis. A total of 318,131 ROH segments were identified in 1,203 individuals across 49 breeds (Appendix A). Populations were categorized into four groups: Asian domestic pigs (ASD), Asian wild boars (ASW), European domestic pigs (EUD), and European wild boars (EUW). European pig populations exhibited significantly higher cumulative ROH length and greater total number of ROH segments per individual compared to Asian counterparts (Figure 1A,B). Population-based ROH analysis revealed distinct patterns between continental groups: Asian populations were predominantly characterized by short ROH segments (0.5–2 Mb), which are indicative of ancient inbreeding events. In contrast, European populations exhibited significantly elevated proportions of both medium (2–5 Mb) and long (>5 Mb) ROH segments (Figure 1C, Appendix A). This pattern strongly suggests intensified recent inbreeding in European populations, likely resulting from modern breeding practices.

### 3.2. Hybridization Reduces ROH Burden

Two hybrid populations, Diannanxiaoer × Duroc (DD) and Duroc × Landrace × Yorkshire (DLY), were examined to assess the effects of crossbreeding. Both exhibited reduced ROH compared to their parental breeds. The DD population, resulting from an Asian–European cross, showed a significant reduction, suggesting that greater parental genetic divergence leads to substantial ROH attenuation (Figure 2A–D). Pairwise F*st* values confirmed that the genetic distance between Diannanxiaoer and Duroc was significantly larger (Figure 2E).

### 3.3. Selection Pressures and Functional Significance of ROH Islands

ROH were non-randomly distributed, with strong selection signals detected within ROH islands. SNPs within ROH island regions exhibited significantly lower π (pi) values and higher Tajima’s D, suggesting reduced genetic diversity and balancing selection. XP-EHH and F*st* analyses further confirmed that European and Asian pigs have experienced distinct selection pressures in ROH regions (Figure 3).

Candidate genes within ROH islands reflected distinct functional roles. The candidate genes in the Asian pig population are significantly associated with growth and immune pathways (Appendix A). In contrast, the candidate genes in the European pig population are predominantly related to taste perception, gonadal hormone secretion, and immune responses to microbial exposure (Figure 4) (Appendix A).

### 3.4. FSTL5: A Key Gene Under Selection in European Pigs

To illustrate the functional significance of genes within ROH islands, we focused on *FSTL5* as a representative example, demonstrating its potential role in shaping phenotypic traits and responding to artificial selection pressures. To investigate the impact of *FSTL5* on pigs, we utilized data from the PigBiobank resource [23]. First, based on phenotype association analysis across 298 traits, we identified *FSTL5* as being associated with the total litter weight of piglets and average daily gain (ADG) (Figure 5A). Second, transcriptome-wide association study (TWAS) results from 34 tissues further confirmed that *FSTL5* is significantly associated with ADG in muscle tissue (Figure 5B). Finally, haplotype analysis of the *FSTL5* gene region in Asian and European pig populations revealed distinct haplotype patterns between the two groups, highlighting potential genetic differentiation at this locus (Figure 5C).

## 4. Discussion

Pigs were domesticated across multiple regions of Eurasia between 9000 and 10,000 years ago, becoming one of the most important livestock species worldwide, supplying more than one-third of the animal protein consumed by humans [24]. Numerous studies have investigated the patterns of ROH in Asian and European pigs separately, exploring their associations with inbreeding and selection [25]. However, due to limitations in sample availability, data consistency, and differences in ROH-calling criteria, the global distribution patterns of ROH across major pig breeds remain largely unexplored. Here, we analyzed high-quality, whole-genome resequencing data from 1203 individuals representing 49 major pig breeds worldwide, revealing striking differences in ROH patterns between European and Asian pig populations. European pigs exhibited a higher number and total length of ROH compared to their Asian counterparts, and hybridization was found to reduce both the number and length of ROH. ROH regions showed strong signatures of selection, with ROH islands in European and Asian pigs enriched for distinct sets of functional genes, reflecting their divergent domestication histories and selection pressures. Notably, our study highlights *FSTL5* as an example of how ROH segments contribute to selection responses by influencing key genes associated with growth and reproduction.

Comparative ROH analysis revealed distinct patterns between Asian and European porcine populations. Asian pigs demonstrated significantly fewer and shorter ROH segments, aligning with archaeological evidence indicating their origins from multiple independent domestication events across East Asia, including distinct lineages in northern China, southern China, and Southeast Asia [26]. In contrast, European domestic pigs principally descended from a single Near Eastern domestication followed by westward dispersal [27], with population bottlenecks explaining their elevated frequency of short ROH fragments. This continental dichotomy was further reflected in European breeding histories: while medieval-initiated systematic selection in commercial breeds has accumulated long ROH segments through recent directional breeding, native European populations (e.g., Dutch heritage breeds: Netherlands) retained predominantly short-to-medium ROH patterns. These observations corroborate two evolutionary trajectories—the singular domestication origin with bottleneck effects (reduced long ROH) versus intensive artificial selection (enhanced long ROH) in European pig. We identified highly inbred individuals within specific Asian populations, such as the Wuzhishan pig. This highlights a potential risk of inbreeding in small populations, underscoring the need for improved breeding management strategies [28]. European pig populations exhibited higher inbreeding coefficients; however, in contrast to purebred parental lines, the hybrid DLY population demonstrated a reduced ROH burden, which aligns with our expectations. This highlights the strategic value of ROH profiling in parental line selection—by identifying purebred individuals with minimized homozygous deleterious segments, breeders can systematically reduce recessive risk alleles in hybrid offspring while retaining heterosis benefits. Our findings necessitate rigorous ROH monitoring within elite purebred stocks as a prerequisite for designing optimal crossbreeding schemes, particularly in pyramid breeding systems where parental genetic quality directly determines commercial herd viability. Furthermore, the near absence of ROH segments in the hybrid DD population suggests extensive historical admixture between Eurasian pig lineages, which likely diverged approximately 1.2 million years ago [29].

ROH islands are non-randomly distributed across the genome and are often associated with selection. Using conventional selection signature detection methods, we confirmed that ROH islands are under significantly stronger selection pressure. Further functional annotation of genes within these ROH regions revealed that both Asian and European pigs showed enrichment for immune-related pathways, underscoring the role of disease resistance as a key selective force across Asian and European pig populations. Functional analyses demonstrated that *CFD*, *AZU1*, and *C3* are significantly enriched in the humoral immune response pathway within European pig populations, where they collectively mediate complement activation and pathogen neutralization [30,31]. In parallel, functional enrichment analysis identified *TRAF7*, *CCNF*, *LZTR1*, *PPIL2*, and *PRMT1* as key regulators in Asian porcine populations, showing prominent enrichment in pathways associated with ubiquitination-mediated degradation, methylation modification, and protein conformational regulation. These genes orchestrate critical biological processes, including immune cell activation, inflammatory signal amplification, and autoimmune homeostasis, highlighting their complementary roles in immune adaptation across geographically distinct pig populations [32,33,34,35]. Additionally, we examined the phenotypic impact of *FSTL5*, one of the most frequent ROH-associated genes in European pigs, and found that it is associated with daily weight gain. The pronounced selection signatures observed at the *FSTL5* locus in European pigs reflect its critical role in enhancing economically vital traits. As a TGF-β superfamily regulator, FSTL5 modulates muscle development and metabolic efficiency—key determinants of growth performance [36,37]. The extended haplotype homozygosity at this locus aligns with centuries of artificial selection in European breeding programs that prioritized daily weight gain and feed conversion efficiency. We identified the *IGFALS* gene within the highest-frequency ROH islands in Asian pig populations. *IGFALS* mediates growth hormone (GH)-insulin-like growth factor-1 (IGF-1) axis signaling to promote tissue growth, a mechanism directly associated with organismal growth rate. This finding aligns with the observed slower growth phenotype prevalent in Asian pig breeds [38,39].

## 5. Conclusions

This study establishes the first global characterization of ROH patterns in pigs, revealing how contrasting Eurasian domestication histories and breeding practices have sculpted genomic architectures. The elevated ROH burden in European breeds versus Asian counterparts underscores the necessity for systematic inbreeding monitoring in purebred stocks, while hybrid vigor effects demonstrated in DLY crosses validate strategic crossbreeding protocols. Prioritizing ROH profiling enables targeted conservation of adaptive genetic variants in indigenous Asian breeds and optimized utilization of European elite lines. Our findings advocate integrating ROH analytics into modern breeding frameworks to balance heterosis exploitation with genetic diversity preservation, thereby enhancing both productivity and resilience in global swine industries.

## Figures and Tables

**Figure 1 animals-15-00988-f001:**
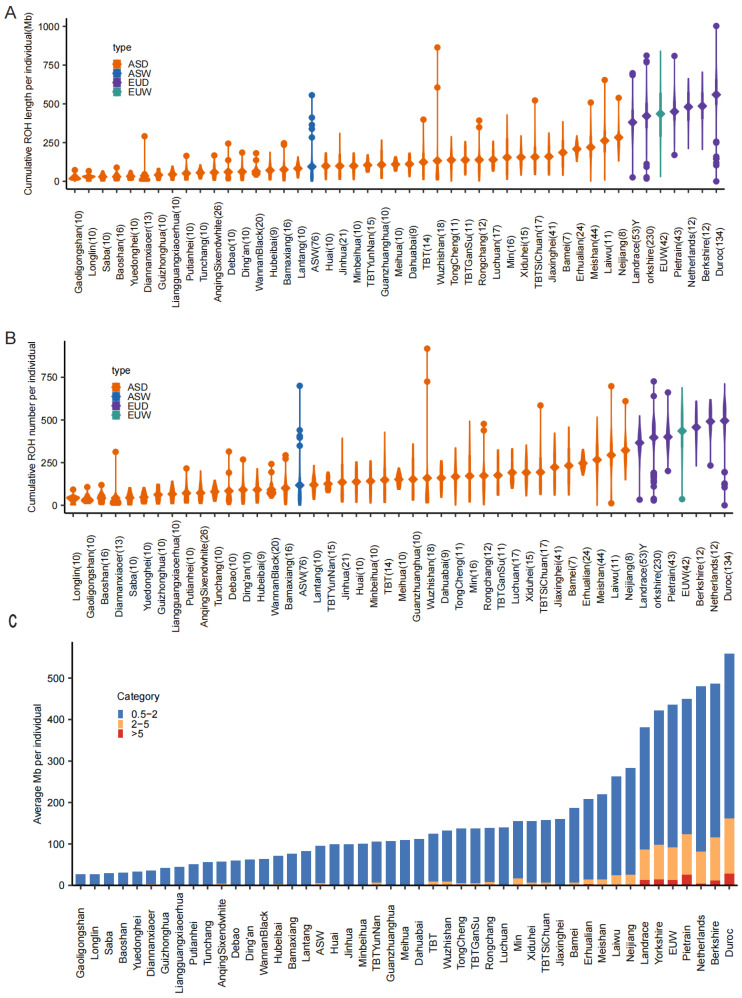
Individual patterns of ROH. The distributions of ROH statistics per individual for 49 pig breeds. We divided the population into four groups (ASD: Asian domestic pigs, ASW: Asian wild boars, EUD: European domestic pigs, EUW: European wild boars) corresponding to the four colors based on geographic location, with the name of the breeds and the number of groups indicated in the horizontal coordinates of the figure. The diamonds indicate the mean value of the population. (**A**) The length of ROH per individual. (**B**) The number of ROH per individual. (**C**) ROH length category distribution. Stacked bar charts depicting the proportion of ROH by length (0.5–2 Mb: short, 2–5 Mb: medium, >5 Mb: long) across the 49 pig breeds, with colors corresponding to ROH categories, and each column representing a breed. Netherlands is a local pig breed originating from the Netherlands. TBT refers to Tibetan pigs from the Qinghai–Tibet Plateau of China. TBTGanSu denotes Tibetan pigs from the Gansu Province, China. TBTSiChuan represents Tibetan pigs from the Sichuan Province, China. TBTYunNan indicates Tibetan pigs from the Yunnan Province, China.

**Figure 2 animals-15-00988-f002:**
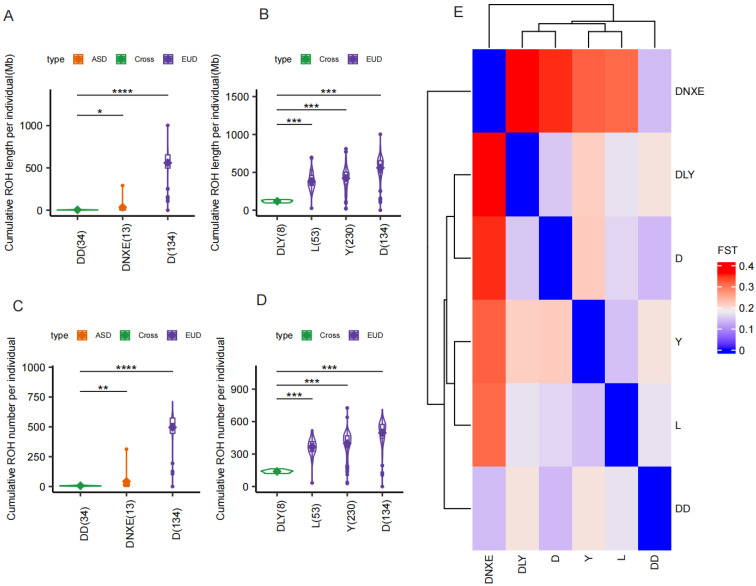
ROH patterns and pairwise F*st* in individuals of hybrid and parental populations. (**A**) Boxplot comparing cumulative ROH length per individual (MB) between the hybrid population DD (Diannanxiaoer × Duroc) and its parental populations (Diannanxiaoer [DNXE] and Duroc [D]). (**B**) Cumulative ROH length comparison for the hybrid population DLY (Duroc × Landrace × Yorkshire) against its parental populations (Duroc [D], Landrace [L], and Yorkshire [Y]). (**C**) Boxplot showing cumulative ROH number per individual in hybrid population DD versus parental populations DNXE and D. (**D**) Cumulative ROH number comparison for hybrid population DLY versus parental populations D, L, and Y. In (**A**–**D**) plots, where diamonds represent mean values, significance levels are indicated by *, **, *** and **** at *p* < 0.05, 0.01, 0.001 and 0.0001, respectively. (**E**) Heatmap of pairwise F*_ST_* values among five populations: Diannanxiaoer (DNXE), Duroc (D), Landrace (L), Yorkshire (Y), and hybrids DD and DLY. Color gradient (blue to red) reflects genetic differentiation intensity, with red indicating higher divergence.

**Figure 3 animals-15-00988-f003:**
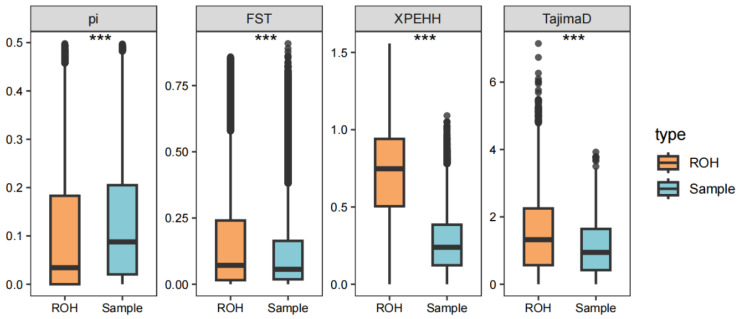
Comparative analysis of selective pressures between ROH-enriched genomic regions and randomly sampled non-ROH island SNPs revealed significantly heightened selection pressures within ROH islands. Significance levels for cohort boxplots were determined using *** for *p* < 0.001.

**Figure 4 animals-15-00988-f004:**
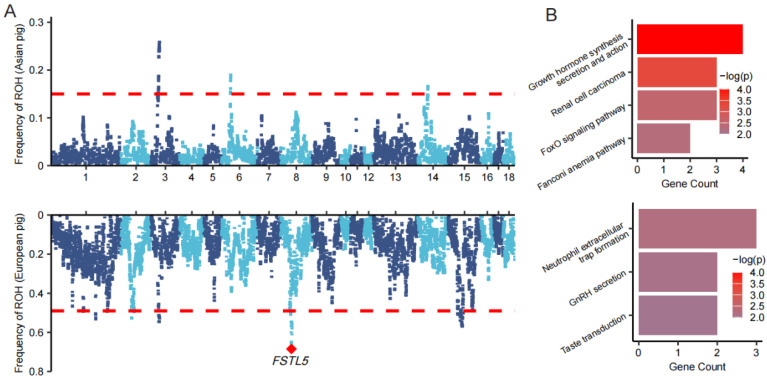
(**A**) Manhattan plot of the distribution of ROH island in the Asian pig (top) and European pig (bottom), where the red line is the top 1% threshold line. (**B**) Significant biological pathways (*p* < 0.01) for GO and KEGG enrichment analysis of candidate genes identified in ROH islands of Asian pig (top) and European pig (bottom) populations.

**Figure 5 animals-15-00988-f005:**
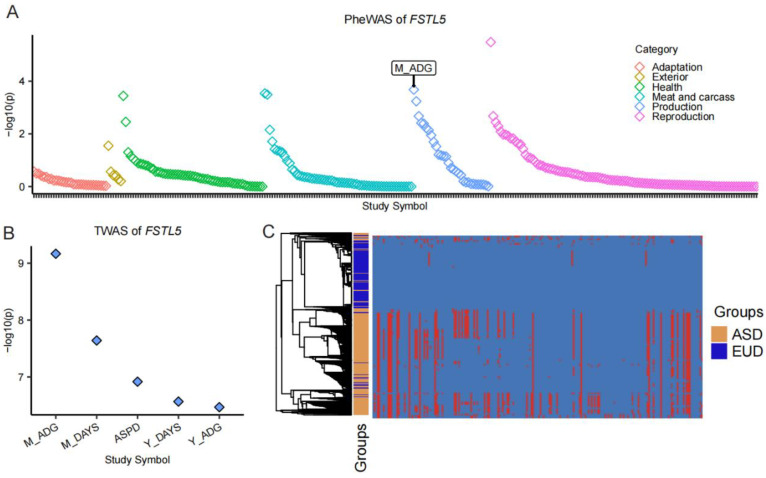
The *FSTL5* gene suffers different selective pressures in European and Asian pig populations. (**A**) The point plot shows the association between *FSTL5* gene and 298 complex traits. (**B**) The point plot shows significant association TWAS results between *FSTL5* gene and traits in muscle tissue. (**A**,**B**) Data obtained from PigBiobank (http://pigbiobank.farmgtex.org, accessed on 11 January 2025). (**C**) Haplotype heat map of Asian and European pigs in the *FSTL5* gene region (50.26–51.05 Mb).

## Data Availability

All the data necessary to evaluate the conclusions of this paper are included within the main manuscript and/or the Appendix A. The 1203 resequenced datasets analyzed in this study were derived from our previously published pig genomics reference panel (PGRP). Detailed information on these datasets is provided in Appendix A.

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
