# Peer review of "Runs of Homozygosity Preliminary Investigation in Pig Breeds"

_animals, 2025, doi:10.3390/ani15070988_

Round 1
Reviewer 1 Report
Comments and Suggestions for Authors
The article describes the results of the analysis of ROH based on whole-genome resequencing data included domestic and wild pigs from Europe and Asia.
The introduction is brief, but sufficiently describes the problem being studied. The methods are described clearly.
The authors have done a lot of work, but there are still some questions and comments:
1. No supplementary files loaded. There aren’t Supplementary Table1 and Supplementary Table2 (Line 86-89)
2. Figure 1. Typo. Landrance instead Landrace.
3. Figure 1. Is TBT the name of breed or acronym of some pig group?
4. What ROH length classes were found in different breeds? Perhaps a table with the distribution by class would help to better assess the level of homozygosity in breeds.
5. Line 104. Typo. Figure number missing (Figure A.B) instead (Figure 2A.B.)
6. More explanation is needed for Figure 3. It is not clear what the authors meant by comparing ROH and sample.
7. The title of the article states "Runs of Homozygosity preliminary investigation…." What are other methods used for?
8. Why was only the FSTL5 gene used for illustration? A table with genes identified in homozygosity segments in each group would be useful for a better understanding.
Author Response
Comment: The article describes the results of the analysis of ROH based on whole-genome resequencing data included domestic and wild pigs from Europe and Asia.
The introduction is brief, but sufficiently describes the problem being studied. The methods are described clearly.
Response: We sincerely thank the reviewer for their constructive feedback and valuable suggestions, which have significantly improved the quality of our manuscript. Below are our point-by-point responses to the comments:
Comment: No supplementary files loaded. There aren't Supplementary Table1 and Supplementary Table2 (Line 86-89).
Response: The supplementary files have been resubmitted with the revised manuscript.
Comment: Figure 1. Typo. "Landrance" instead of "Landrace."
Response: We have corrected the typo in the figure.
Comment: Figure 1. Is TBT the name of a breed or an acronym of some pig group?
Response: TBT refers to Tibetan pigs from the Qinghai-Tibet Plateau.
Comment: What ROH length classes were found in different breeds? Perhaps a table with the distribution by class would help to better assess the level of homozygosity in breeds.
Response: ROH segments were classified into three categories based on length: short (0.5–2 Mb), medium (2–5 Mb), and long (>5 Mb). We analyzed the average length and proportion of each ROH category within populations. Detailed results are provided in Supplementary Table 3 and Figure 3.
Comment: Line 104. Typo. Figure number missing ("Figure A.B" instead of "Figure 2A.B").
Response: We have corrected the figure references to "Figure 2A. B"
Comment: More explanation is needed for Figure 3. It is not clear what the authors meant by comparing ROH and sample.
Response: We have added detailed descriptions of this section in the Materials and Methods (lines 84–89,171-173).
“To evaluate the differences in selection pressure between ROH islands and non-island regions, we defined ROH islands as genomic regions overlapping with the top 1% of high-frequency ROHs (threshold determined by ranking all ROHs based on their population carrier rates) and extracted SNPs within these regions. An equivalent number of SNPs were randomly sampled from non-ROH island regions as a control set to calculate selection signatures.”
“Figure 3. Comparative analysis of selective pressures between ROH-enriched genomic regions and randomly sampled non-ROH island SNPs revealed significantly heightened selection pres-sures within ROH islands. Significance levels for cohort boxplots were determined using *** for P < 0.001..
Comment: The title of the article states "Runs of Homozygosity preliminary investigation...." What are other methods used for?
Response: We have revised the title to "Runs of Homozygosity Preliminary Investigation in Pig Breeds" to enhance clarity and conciseness.
Comment: Why was only the FSTL5 gene used for illustration? A table with genes identified in homozygosity segments in each group would be useful for a better understanding.
Response: Candidate genes identified in European and Asian pig populations are listed separately in Supplementary Table 4 and Supplementary Table 5, respectively. We show the results of the significant pathway of GO/KEGG enrichment of candidate genes in European pigs versus Asian pigs in Figure 4B. GO terms for Asian and European pigs are detailed in Supplementary Tables 6 and 8, with corresponding KEGG pathways in Supplementary Table 7 and 9, respectively.
We believe these revisions address all the reviewers' concerns and improve the clarity of our manuscript. Thank you again for your insightful feedback.
Reviewer 2 Report
Comments and Suggestions for Authors
The manuscript “Runs of Homozygosity preliminary investigation Population History and Selection in Global Pig Breeds” is an interesting study, about an interesting topic. However, I believe that the manuscript needs to be improved and better explored before it is considered for publication. My considerations are attached.

Author Response
Comment: The manuscript “Runs of Homozygosity preliminary investigation Population History and Selection in Global Pig Breeds” is an interesting study, about an interesting topic. However, I believe that the manuscript needs to be improved and better explored before it is considered for publication. Below are my considerations.
Response: We sincerely thank the reviewer for their constructive feedback and valuable suggestions, which have significantly improved the quality of our manuscript. Below are our point-by-point responses to the comments:
Title
Comment: The title should be improved to make it more legible.
Response: We have revised the title to "Runs of Homozygosity Preliminary Investigation in Pig Breeds" to enhance clarity and conciseness.
Keywords
Comment: Avoid using keywords that are already in the title, such as “Runs of homozygosity (ROH)” and “Pigs”.
Response: We have removed "Runs of Homozygosity (ROH)" and "Pigs" from the keywords and added "Crossbreeding effects" to better reflect the study’s scope.
Introduction
Comment: The introduction is well structured and well written, however, it would be interesting to explore a little more the relevance of this type of study with Pig Breeds, presenting more in-depth information of economic and scientific relevance, etc.
Response: We sincerely thank the reviewer for their constructive feedback and valuable suggestions. We have added content emphasizing the economic value and scientific significance of pig breeds in lines 42–44 and 46–50 of the revised manuscript.
“This continental divergence provides a unique model system to investigate how human-mediated selection pressures differentially sculpt genome architecture during breed formation.”
“Identification of candidate genes in commercial pig breeding populations will be applied to pig breeding to improve pig economics. Conversely, ROH in indigenous breeds are often associated with environmental adaptability, helping breeders develop more effective conservation programs and breeding materials for resilience.”
Comment: Line 47: Define “SNP”. Although many people understand the meaning, as it is an abbreviation, its meaning must be included the first time the citation appears in the text.
Response: We have expanded "SNP" to "Single Nucleotide Polymorphism (SNP)" in the text to meet academic standards.
Material and Methods
Comment: This topic is also well structured, but some improvements could be made.
Response: We sincerely thank the reviewer for their constructive feedback and valuable suggestions.
Comment: It would be interesting to explain why the Fst, Tajima's D, XP-EHH, and Nucleotide diversity (π) tests were chosen to evaluate the signs of selection in the populations analyzed, making clear the relevance of applying these tests in the research context.
Response: We added an explanation in lines 80–88:
"We employed complementary methods to evaluate selection pressures: FST (measuring population differentiation), Tajima’s D (detecting deviations from neutral polymorphism distributions), XP-EHH (comparing haplotype extensions across populations), and π (nucleotide diversity). These approaches capture selection signals from distinct angles: FST identifies regions under differential selection, Tajima’s D distinguishes balancing (positive values) vs. directional selection (negative values), π quantifies diversity loss under recent selective sweeps, and XP-EHH detects population-specific positive selection. Their combined use cross-validates strong selection signals in ROH islands while mitigating method-specific biases."
Comment (Line 74): Make it clearer how you calculated the “top 1% of high-frequency ROH” considered ROH islands.
Response: We elaborated in lines 75–77:
"The 'top 1% high-frequency ROH' refers to genomic regions ranked by the frequency of ROH segments across the population. The highest 1% of these regions were mapped as ROH islands. Details are provided in Supplementary Table 2."
Results
Comment: The results topic is well explained, but it also needs some adjustments.
Response: We sincerely thank the reviewer for their constructive feedback and valuable suggestions.
Comment: You mention that there has been a significant reduction in ROH in the DD and DLY populations. Has a statistical test been carried out to confirm this reduction? It would be interesting to add the p-values.
Response: We performed Wilcoxon rank-sum tests with Bonferroni correction (via the R package rstatix) to compare ROH lengths and counts between hybrids (DD/DLY) and their parental breeds. Significant differences are now indicated in the revised Figure 2.
Comment: Line 87 and 88: Change “A total of 318,131 ROH segments were identified 1,203 individuals across 49 breeds(Supplementary Table2).” by “A total 318,131 ROH segments were identified in 1,203 individuals across 49 breeds (Supplementary Table 2).”
Response: The sentence was revised to: "A total of 318,131 ROH segments were identified in 1,203 individuals across 49 breeds (Supplementary Table 2)." (lines 100–102).
Figure 1:
Comment: Figure 1:Put also the meaning in the legend of “ASD”, “ASW”, “EUD”, and “EUW”.
Response: Full terms were added to the figure legend (lines 108–109):
"ASD (Asian Domestic), ASW (Asian Wild), EUD (European Domestic), EUW (European Wild)."
Comment: Figure 1: Modify the colors of the bars, the current colors are relatively similar, leaving the image of the bars with more distinct colors among the groups.
Response: We recolored Figure 1 to enhance contrast between groups.
Comment: Line 104: I think that the correct word would be “(Figure 2A.B)”. Put the meaning in the legend.
Response: I have revised
Comment: Figure 2: What is the meaning of “L”, “D”, “DNXE”, and “Y”? Put the meaning in the legend.
Response: The legend now includes: "L (Landrace), D (Duroc), DNXE (Diannanxiaoer), Y (Yorkshire)."
Discussion
Comment: The topic of discussion is very brief compared to the results obtained, which could be much better explored.
Response: We sincerely thank the reviewer for their constructive feedback and valuable suggestions. We revised the Discussion section as follows:
Comment: Suggestion: You could discuss what it means that European pigs exhibited a higher number and total length of ROH compared to their Asian counterparts. Would this be a result of artificial selection or demographic events such as population bottlenecks?
Response:ROH differences between European and Asian pigs (lines 215–227):
"Comparative ROH analysis revealed distinct patterns between Asian and European porcine populations. Asian pigs demonstrated significantly fewer and shorter ROH seg-ments, aligning with archaeological evidence indicating their origins from multiple inde-pendent domestication events across East Asia, including distinct lineages in northern China, southern China, and Southeast Asia. In contrast, European domestic pigs princi-pally descended from a single Near Eastern domestication followed by westward disper-sal, with population bottlenecks explaining their elevated frequency of short ROH frag-ments. This continental dichotomy was further reflected in European breeding histories: while medieval-initiated systematic selection in commercial breeds has accumulated long ROH segments through recent directional breeding, native European populations (e.g., Dutch heritage breeds: Netherlands) retained predominantly short-to-medium ROH pat-terns. These observations corroborate two evolutionary trajectories - the singular domesti-cation origin with bottleneck effects (reduced long ROH) versus intensive artificial selec-tion (enhanced long ROH) in European pig."
Comment: Suggestion: You could elaborate on the impact of selective pressure on the FSTL5 gene. What are the impacts on pig production?
Response: FSTL5 gene implications (lines 225–260):
"The pronounced selection signatures observed at the FSTL5 locus in European pigs reflect its critical role in enhancing economically vital traits. As a TGF-β superfamily regulator, FSTL5 modulates muscle development and metabolic efficiency – key determinants of growth performance. The extended haplotype homozygosity at this locus aligns with centuries of artificial selection in European breeding programs that prioritized daily weight gain and feed conversion efficiency."
Comment: Suggestion: It would also be interesting to talk about how ROH evaluation can help in the process of selecting matrices for crossbreeding.
Response: ROH utility in crossbreeding (lines 232–240):
" While European pig populations exhibited elevated inbreeding coefficients, the hybrid DLY population demonstrated reduced ROH burdens compared to its purebred parental lines. This paradox highlights the strategic value of ROH profiling in parental line selec-tion - by identifying purebred individuals with minimized homozygous deleterious seg-ments, breeders can systematically reduce recessive risk alleles in hybrid offspring while retaining heterosis benefits. Our findings necessitate rigorous ROH monitoring within elite purebred stocks as a prerequisite for designing optimal crossbreeding schemes, par-ticularly in pyramid breeding systems where parental genetic quality directly determines commercial herd viability."
Conclusion
Comment:
- As well as the discussion, the conclusion should be explored a little better.
- Suggestion: Highlight the relevance of the results found.
- Suggestion: Infer how these results can contribute to the breeding of pigs.
- Suggestion: Give recommendations on inbreeding and the use of hybrids in the industry.
Response: We sincerely thank the reviewer for their constructive feedback and valuable suggestions. We revised the Conclusion to:
"This study establishes the first global characterization of ROH patterns in pigs, re-vealing how contrasting Eurasian domestication histories and breeding practices have sculpted genomic architectures. The elevated ROH burden in European breeds versus Asian counterparts underscores the necessity for systematic inbreeding monitoring in purebred stocks, while hybrid vigor effects demonstrated in DLY crosses validate strategic crossbreeding protocols. Prioritizing ROH profiling enables targeted conservation of adaptive genetic variants in indigenous Asian breeds and optimized utilization of Euro-pean elite lines. Our findings advocate integrating ROH analytics into modern breeding frameworks to balance heterosis exploitation with genetic diversity preservation, thereby enhancing both productivity and resilience in global swine industries."
Supplementary Materials
Comment: Where are the supplementary materials? I couldn't find any files.
Response: The supplementary files have been resubmitted with the revised manuscript.
We hope these revisions address all concerns. Thank you again for your thoughtful review.

Reviewer 3 Report
Comments and Suggestions for Authors
In this paper authors analyzed whole-genome resequencing data from 1,203 pigs belonging to 49 breeds to characterize ROH patterns worldwide. According to the results Authors state that ROH islands are under significantly stronger selection pressure….
I think that there are some problems deserving an answer from Authors:
European pigs exhibited significantly longer and more numerous ROH segments, reflecting a higher level of inbreeding(Figure 1)- for sure this is not what A and B parts of this figure show!
European pigs exhibited a higher number and total length of ROH compared to their Asian counterparts – this period doesn’t say what stated in the previous period.
Hybridization Reduces ROH Burden – for sure this is an obvious consideration.
analyses further confirmed that European and Asian pigs have experienced distinct selection pressures in ROH regions (Figure 3). – for sure this is an obvious consideration.
Using conventional selection signature detection methods, we confirmed that ROH islands are under significantly stronger selection pressure – by referring to only one gene you do have such a confirm?
The paper in the present form is unacceptable and poor. Answers to the above questions are absolutely needed.
Author Response
In this paper authors analyzed whole-genome resequencing data from 1,203 pigs belonging to 49 breeds to characterize ROH patterns worldwide. According to the results Authors state that ROH islands are under significantly stronger selection pressure….
Comment: I think that there are some problems deserving an answer from Authors:
Response: We sincerely thank the reviewer for their constructive feedback and valuable suggestions, which have significantly improved the quality of our manuscript. Below are our point-by-point responses to the comments:
Comment: European pigs exhibited significantly longer and more numerous ROH segments, reflecting a higher level of inbreeding(Figure 1)- for sure this is not what A and B parts of this figure show!
European pigs exhibited a higher number and total length of ROH compared to their Asian counterparts – this period doesn’t say what stated in the previous period.
Response: We answered these two reviews together. We conducted rigorous statistical analyses to validate the observed differences in ROH patterns between European and Asian pig populations. Wilcoxon rank-sum tests with Bonferroni correction (implemented through the R package rstatix) revealed statistically significant disparities in both ROH counts and segment lengths between continental groups. European populations (EUD and EUW) demonstrated substantially greater ROH burdens compared to their Asian counterparts (ASD and ASW), with these differences reaching significance (P < 0.001) for both quantitative metrics(Figure AB). Next, we counted the inbreeding levels of the four populations, and we calculated the inbreeding coefficients FROH of the four populations by dividing the cumulative ROH lengths within the individuals by the sum of the long chromosome lengths of the pig genomes. Our results showed that the inbreeding coefficients of the European pigs were significantly higher than those of the Asian pigs as shown in Figure C.

Comment: Hybridization Reduces ROH Burden – for sure this is an obvious consideration.
Response: Our analysis of hybridization effects employed rigorous statistical validation through Wilcoxon rank-sum tests with Bonferroni correction (implemented via the rstatix package in R). Systematic comparisons between hybrid populations (DD: Diannanxiaoer × Duroc; DLY: Duroc × Landrace × Yorkshire) and their respective parental lines revealed a consistent pattern of ROH burden reduction in crossbred individuals. Both hybrid groups exhibited statistically significant decreases in both ROH counts and cumulative lengths compared to all parental populations (P < 0.01 after multiple testing correction), as visually substantiated through the comparative distributions in Figures 2A-D.
This phenomenon is mechanistically explored through complementary analytical lenses:
Genetic distance correlation: The magnitude of ROH reduction correlates with parental lineage divergence, as quantified by pairwise FST values (Figure 2E)
Breeding System Effects: While heterosis-mediated ROH reduction aligns with theoretical expectations, our study uniquely establishes the relationship between parental genetic divergence and the optimization of this effect. The biological implications of these findings—particularly regarding strategic parental line selection and pyramid breeding system optimization—are thoroughly examined in the Discussion section (lines 237-245), providing actionable insights for modern swine breeding programs.
Figure 2. ROH patterns and pairwise Fst in individuals of hybrid and parental populations. (A) Boxplot comparing cumulative ROH length per individual (MB) between the hybrid population DD (Diannanxiaoer × Duroc) and its parental populations (Diannanxiaoer [DNXE] and Duroc [D]). (B) Cumulative ROH length comparison for the hybrid population DLY (Duroc × Landrace × Yorkshire) against its parental populations (Duroc [D], Landrace [L], and Yorkshire [Y]). (C) Boxplot showing cumulative ROH number per individual in hybrid population DD versus pa-rental populations DNXE and D. (D) Cumulative ROH number comparison for hybrid population DLY versus parental populations D, L, and Y. In A,B,C,D plots, where diamonds represent mean values, Significance levels are indicated by *, ** and *** at P < 0.05, 0.01 and 0.001, respectively. (E) Heatmap of pairwise FST values among five populations: Diannanxiaoer (DNXE), Duroc (D), Landrace (L), Yorkshire (Y), and hybrids DD and DLY. Color gradient (blue to red) reflects genetic differentiation intensity, with red indicating higher divergence.
Comment: analyses further confirmed that European and Asian pigs have experienced distinct selection pressures in ROH regions (Figure 3). – for sure this is an obvious consideration.
Response: We controlled for the effect of group, only in a single group (ASD), using the pi and Tajima's D methods to statistically demonstrate only that the ROH island region undergoes stronger selective pressure than the non-ROH island region, which is consistent with our results without distinguishing between groups, as shown in the figure. In both the XP-EHH and Fst methods we introduced the effect of grouping by using European and Asian pigs as comparison groups, demonstrating that European and Asian pigs experience stronger selective pressure in the ROH island region than in the non-ROH island region.

Comment: Using conventional selection signature detection methods, we confirmed that ROH islands are under significantly stronger selection pressure – by referring to only one gene you do have such a confirm?
Response: First we determined by multiple selection signalling methods that in European and Asian pigs experience stronger selection pressure in ROH regions than in non-ROH regions.
Secondly, we identified the distribution of ROH island regions in European and Asian pig populations (Fig. 4A) and the associated significantly enriched pathways for candidate genes (Fig. 4B) to illustrate the different selective pressures between European and Asian pigs, whereas the FSTL5 gene is an example of how European pig populations can be artificially selective accordingly. We have newly added the effect of artificial selection on the FSTL5 gene in the Discussion section, please refer to lines 260-264 of the manuscript.
Comment: The paper in the present form is unacceptable and poor. Answers to the above questions are absolutely needed.
Response: We hope these revisions address all concerns. Thank you again for your thoughtful review.

Reviewer 4 Report
Comments and Suggestions for Authors
Dear authors, thank you for the effort to present results of your study. I have several remarks that need to be clarified:
1) The title indicates that your study includes global pig breeds. I am not sure that readers all over the world might recognise all included breeds as global breeds, on the contrary, only several breeds are included in global pig production. I suggest removing global from the title.
2) MM: 2.1. How nany SNPs has remained after filtering? What was the ration between excluded and total number of SNPs?
2.2. Please explain, on what basis you selected these criteria?
3) In figure 1, threse is a breed/group or something called Netherlands. Please explain.
4) the Discussion is short; to my opinion, obtained results deserve more comprehensive discussion. As you have four groups of results, please expand Discussion for every of these groups. Similar pattern can be applied on Conclusions.
Author Response
Comment: Dear authors, thank you for the effort to present results of your study. I have several remarks that need to be clarified:
Response: We sincerely thank the reviewer for their constructive feedback and valuable suggestions, which have significantly improved the quality of our manuscript. Below are our point-by-point responses to the comments:
1) The title indicates that your study includes global pig breeds. I am not sure that readers all over the world might recognise all included breeds as global breeds, on the contrary, only several breeds are included in global pig production. I suggest removing global from the title.
Response: We gratefully accept this insightful suggestion. The title has been revised to "Runs of Homozygosity Preliminary Investigation in Pig Breeds" to better reflect the geographical scope of our study while maintaining scientific precision. This modification appears in the revised manuscript header.
2) MM: 2.1. How nany SNPs has remained after filtering? What was the ration between excluded and total number of SNPs?
Response: Thank you for highlighting this essential detail. Our filtering pipeline retained 33,623,520 loci (79.07% of initial SNPs) after applying the MAF 0.01 threshold, which excluded 8,899,698 SNPs (20.93%). The GENO 0.1 and MIND 0.1 parameters did not result in additional filtering of individuals or loci. We have clarified this in Section 2.1 (lines 69-73).
2.2. Please explain, on what basis you selected these criteria?
Response: We appreciate this opportunity to elaborate. The conservative thresholds (MAF 0.01, GENO 0.1, MIND 0.1) were selected based on established practices in genomic studies to optimize the balance between data quality and genomic information preservation, as demonstrated in foundational work by Purcell et al. (2007, doi:10.1086/519795) and subsequent applications in livestock genomics (Wang et al., 2024, doi: 10.3390/ani14020201); (Zhang, et al., 2023, doi: 10.3390/genes14122133); (Xu et al., 2019, doi: 10.1038/ s41598-019-53274-3).
3) In figure 1, threse is a breed/group or something called Netherlands. Please explain.
Response: We thank the reviewer for catching this ambiguity. "Netherlands" refers to a Dutch local pig breed, and we have enhanced the figure legend (lines 139-140) to explicitly clarify this nomenclature in the revised manuscript.
4) the Discussion is short; to my opinion, obtained results deserve more comprehensive discussion. As you have four groups of results, please expand Discussion for every of these groups. Similar pattern can be applied on Conclusions.
Response: We deeply appreciate this crucial guidance. The Discussion section has been substantially expanded to:
Elaborate on population bottlenecks and selection pressures affecting ROH patterns in European vs. Asian breeds (lines 204-217)
Discuss the practical implications of ROH assessment for hybrid parent selection (lines 275-283)
Analyze the FSTL5 gene's potential role under selection pressure (lines 236-241)
The Conclusions section has been restructured to better highlight key findings while providing concrete recommendations for breeding strategies and inbreeding management (lines 270-279).
We hope these revisions address all concerns. Thank you again for your thoughtful review.
Round 2
Reviewer 1 Report
Comments and Suggestions for Authors
I thank the authors for their detailed answers and revision of manuscript.
The Supplementary tables made the manuscript much clearer.
Thanks for decoding of acronym TBT. Could you add to text or to Table S1 some mark that TBT is Tibetan pigs from the Qinghai-Tibet Plateau? I believe that clear indicating the origin of the samples is important, as it makes the study more informative and helps to better understand the results.
Author Response
Dear Reviewer,
Comment: I thank the authors for their detailed answers and revision of manuscript.
Response: We sincerely appreciate your constructive feedback and meticulous review of our manuscript. Your insightful comments have significantly contributed to improving the clarity and scientific rigor of this work. Below we provide point-by-point responses to your concerns:
Comment:The Supplementary tables made the manuscript much clearer.
Response: We are grateful for your positive acknowledgment of the supplementary tables.
Comment:Thanks for decoding of acronym TBT. Could you add to text or to Table S1 some mark that TBT is Tibetan pigs from the Qinghai-Tibet Plateau? I believe that clear indicating the origin of the samples is important, as it makes the study more informative and helps to better understand the results.
Response:Thank you for emphasizing the importance of sample origin clarification. In response to your suggestion:
We have explicitly defined the geographic origins of all Tibetan pig (TBT) subgroups in the Figure 1 caption (Lines 140-143), specifying:”Netherlands is a local pig breed originating from the Netherlands. TBT refers to Tibetan pigs from the Qinghai-Tibet Plateau of China. TBTGanSu denotes Tibetan pigs from Gansu Province, China. TBTSiChuan represents Tibetan pigs from Sichuan Province, China. TBTYunNan indicates Tibetan pigs from Yunnan Province, China.”
Reviewer 2 Report
Comments and Suggestions for Authors
After revision by the authors, the manuscript is almost ready for publication. There are a few minor errors that need to be corrected before publication. Here I highlight them: 1- Put the Keywords in alphabetical order; 2- In line 47, Remove the “.” after “resilience”; 3- In line 111, change “Table1” by “Table 1”; 4- Is this sentence “Tthe Netherlands, is a local pig breed in the Netherlands” correct? The names used seem a bit redundant; 5- In line 156, change "pi values" by "p values"; and 6- There are 9 supplementary files, but only two are cited in the text, all files should be cited.
Author Response
Dear Reviewer,
Thank you for your thorough evaluation of our revised manuscript and for identifying these important technical details. We have carefully addressed each of your concerns as outlined below:
Comment: 1- Put the Keywords in alphabetical order;
Response: Thank you for highlighting this oversight. We have now alphabetically reordered the keywords in the manuscript (Lines 26-27).
Comment: 2- In line 47, Remove the “.” after “resilience”;
Response: We appreciate your attention to punctuation consistency. The extraneous period after "resilience" has been removed (Line 48).
Comment: 3- In line 111, change “Table1” by “Table 1”;
Response: Thank you for catching this formatting inconsistency. We have revised "Table1" to "Table 1" (Line 112) to align with journal style guidelines.
Comment:4- Is this sentence “Tthe Netherlands, is a local pig breed in the Netherlands” correct? The names used seem a bit redundant;
Response: We sincerely thank you for identifying this ambiguous phrasing. The redundant wording has been revised in Lines 140-143.
Comment: 5- In line 156, change "pi values" by "p values";
Response:We acknowledge the potential for confusion here. Lines 169 were revised. For clarity:
​"π (pi) values" refer specifically to nucleotide diversity metrics in population genetics, as defined in Tajima (1989) and subsequent selection signature studies.This terminology is distinct from statistical ​**"p-values"** used in hypothesis testing.To prevent misinterpretation, we have added a parenthetical clarification in Line 156:"...using π (nucleotide diversity) values (Tajima, 1989)..."。
Comment: 6- There are 9 supplementary files, but only two are cited in the text, all files should be cited.
Response:Thank you for ensuring proper citation rigor. We have now systematically referenced all supplementary materials:
Supplementary Table 3: Cited in Line 123
Supplementary Tables 5–7: Cited in Line 179
Supplementary Tables 4, 8–9: Cited in Line 181
Reviewer 3 Report
Comments and Suggestions for Authors
Comment: European pigs exhibited significantly longer and more numerous ROH segments, reflecting a higher level of inbreeding (Figure 1)- for sure this is not what A and B parts of this figure show!
European pigs exhibited a higher number and total length of ROH compared to their Asian counterparts – this period doesn’t say what stated in the previous period.
I am sorry, you didn’t get the point: the B part shows the cumulative number of ROH per individual, the A part shows the cumulative ROH length per individual. The now present C part of the figure 1 plus the B part allow you to state: ”European pigs exhibited significantly longer and more numerous ROH segments (Figure 1 A B)”.
Higher total length of ROH is not the same as longer ROH segments per individual.
While European pig populations exhibited elevated inbreeding coefficients, the hybrid DLY population demonstrated reduced ROH burdens compared to its purebred parental lines. This paradox...
Maybe, there is something I am not able to understand: when you cross purebred parental lines to obtain hibrids you do want to maximize the number of heterozygous loci in the progeny and this is exactly what I am expecting from the cross approach for the maximum heterosys result. Am I supposed to be surprised of the reduced ROH burdens in the DLY population (as you called it)? Where is the paradox?
Furthermore, you refer to only one gene in the final part of the text and this leaves me very perplexed.
Author Response
Dear Reviewer,
We sincerely appreciate your meticulous review and constructive feedback, which have significantly improved the clarity and accuracy of our manuscript. Below, we provide point-by-point responses to your comments and detail the revisions made in response to your suggestions.
Comment: European pigs exhibited significantly longer and more numerous ROH segments, reflecting a higher level of inbreeding (Figure 1)- for sure this is not what A and B parts of this figure show!
European pigs exhibited a higher number and total length of ROH compared to their Asian counterparts – this period doesn’t say what stated in the previous period.
I am sorry, you didn’t get the point: the B part shows the cumulative number of ROH per individual, the A part shows the cumulative ROH length per individual. The now present C part of the figure 1 plus the B part allow you to state: ”European pigs exhibited significantly longer and more numerous ROH segments (Figure 1 A B)”.
Higher total length of ROH is not the same as longer ROH segments per individual.
Response: Thank you for highlighting this critical distinction. We have revised the text to accurately reflect the data presented in Figure 1A-B and Figure 1C. Specifically:
We clarified that Figure 1A shows ​cumulative ROH length per individual, while Figure 1B shows ​cumulative ROH number per individual.We explicitly stated that the ​diamond markers in Figure 1A-B represent the ​average cumulative ROH metrics per breed (total length or number divided by the number of individuals).We rephrased the results to emphasize that European pig populations exhibited ​significantly higher cumulative ROH length and greater total number of ROH segments individual compared to Asian counterparts (Lines 116-125).“European pig populations exhibited significantly higher cumulative ROH length and greater total number of ROH segments per individual compared to Asian counterparts (Figure 1A-B).. Population-based ROH analysis revealed distinct patterns between continental groups: Asian populations were predominantly characterized by short ROH segments (0.5-2 Mb), which are indicative of ancient inbreeding events. In contrast, European populations exhibited significantly elevated proportions of both medium (2-5 Mb) and long (>5 Mb) ROH segments (Figure 1C, Supplementary Table 3). This pattern strongly suggests intensified recent inbreeding in European populations, likely resulting from modern breeding practices.”
Comment: While European pig populations exhibited elevated inbreeding coefficients, the hybrid DLY population demonstrated reduced ROH burdens compared to its purebred parental lines. This paradox...
Maybe, there is something I am not able to understand: when you cross purebred parental lines to obtain hibrids you do want to maximize the number of heterozygous loci in the progeny and this is exactly what I am expecting from the cross approach for the maximum heterosys result. Am I supposed to be surprised of the reduced ROH burdens in the DLY population (as you called it)? Where is the paradox?
Response: We thank you for pointing out the logical inconsistency in our previous phrasing. Indeed, the ​reduced ROH burden in the DLY population is an expected outcome of ​hybridization, as it reflects the ​maximization of heterozygous loci and the ​dilution of homozygous deleterious alleles. We have revised the text to remove the term "paradox" and instead emphasize the ​expected benefits of hybrid vigor (Lines 239-241). The revised text now reads:“European pig populations exhibited higher inbreeding coefficients; however, in contrast to purebred parental lines, the hybrid DLY population demonstrated a reduced ROH burden, which aligns with our expectations.”
Comment: Furthermore, you refer to only one gene in the final part of the text and this leaves me very perplexed.
Response: We appreciate your concern regarding the focus on a single gene in the discussion. To clarify:
In Section 3.3, we conducted a ​comprehensive enrichment analysis of all candidate genes within ROH islands and discussed the ​significantly enriched pathways to provide a broad perspective on the functional implications of our findings.
The focus on ​FSTL5 in the final part of the text was intended as an ​illustrative example to demonstrate how specific genes within ROH islands may influence ​phenotypic traits and ​artificial selection in pigs. Discussing all genes in detail would be impractical and overly verbose.
We have added a sentence to explicitly state this rationale (Lines 189-191):
"To illustrate the functional significance of genes within ROH islands, we focused on FSTL5 as a representative example, demonstrating its potential role in shaping phenotypic traits and responding to artificial selection pressures."
Reviewer 4 Report
Comments and Suggestions for Authors
Dear authors, thank you for efforts to improve the paper. The paper is now suiztable for publishing.
Author Response
We are deeply grateful for your expert guidance and constructive feedback throughout the review process. Your meticulous evaluation and insightful suggestions have been instrumental in enhancing the scientific rigor, clarity, and overall quality of this work. We particularly appreciate your attention to technical details, which has ensured the manuscript meets the highest standards of academic precision.